# Influence of Public Oral Health Services and Socioeconomic Indicators on the Frequency of Hospitalization and Deaths due to Oral Cancer in Brazil, between 2002–2017

**DOI:** 10.3390/ijerph18010238

**Published:** 2020-12-31

**Authors:** Aldelany R. Freire, Deborah E. W. G. Freire, Elza C. F. de Araújo, Edson H. G. de Lucena, Yuri W. Cavalcanti

**Affiliations:** Graduate Program in Dentistry, Department of Clinic and Social Dentistry, Federal University of Paraíba, University City, João Pessoa-PB 58046-600, Brazil; aldelany.ramalho@academico.ufpb.br (A.R.F.); ellenwg.d@gmail.com (D.E.W.G.F.); ecfaraujo@hotmail.com (E.C.F.d.A.); ehglucena@gmail.com (E.H.G.d.L.)

**Keywords:** oral cancer, socioeconomic factors, health services coverage, primary health care, healthcare disparities, health systems

## Abstract

*Background*: Oral cancer is a frequent neoplasm worldwide, and socioeconomic factors and access to health services may be associated with its risk. Aim: To analyze effect of socioeconomic variables and the influence of public oral health services availability on the frequency of new hospitalized cases and mortality of oral cancer in Brazil. *Materials and Methods*: This observational study analyzed all Brazilian cities with at least one hospitalized case of oral cancer in the National Cancer Institute database (2002–2017). For each city were collected: population size, Municipal Human Development Index (MHDI), Gini Coefficient, oral health coverage in primary care, number of Dental Specialized Centers (DSC) and absolute frequency of deaths after one year of the first treatment. The risk ratio was determined by COX regression, and the effect of the predictor variables on the incidence of cases was verified by the Hazard Ratio measure. Poisson regression was used to determine factors associated with higher mortality frequency. *Results*: Cities above 50,000 inhabitants, with high or very high MHDI, more unequal (Gini > 0.4), with less oral health coverage in primary care (<50%) and without DSC had a greater accumulated risk of having 1 or more cases (*p* < 0.001). Higher frequency of deaths was also associated with higher population size, higher MHDI, higher Gini and lower oral health coverage in primary care (*p* < 0.001). *Conclusions*: The number hospitalization and deaths due to oral cancer in Brazil was influenced by the cities’ population size, the population’s socioeconomic status and the availability of public dental services.

## 1. Introduction

Oral cancer represents one of the most frequent neoplasms worldwide, with greater prominence in developing countries, although the incidence varies between different regions [1,2]. In Brazil, 10.70 new cases are estimated for every 100,000 men and 3.71 for every 100,000 women, in each year of the 2020–2022 triennium [3]. According to this estimate, the disease will represent the fifth most common type of cancer among men, and the thirteenth among women by 2022 [3].

Smoking and alcoholism are considered the main risk factors for oral cancer [4,5]. Other aspects such as human papillomavirus infection, solar radiation and genetics are also part of its complex etiology [2,5,6]. In addition, socioeconomic factors are strongly associated with the risk of oral cancer [7,8]. In general, individuals with worse socioeconomic status are more exposed to behavioral risk factors (such as tobacco and alcohol consumption) and have limited access to health services throughout their lives, which contributes to less prevention and late diagnosis of oral cancer [9,10].

In Brazil, the public oral health services coverage has undergone a marked expansion since 2004, with the implementation of the National Oral Health Policy (NOHP) [11]. This policy provided an increase in access to oral health in primary care, in addition to implementing specialized dental care services throughout the country, through the creation of the Dental Specialized Centers (DSC) [12]. NOHP prioritized oral cancer care at different levels of care, fostering prevention actions, active screening, and early diagnosis [12,13].

Since Brazil is a country marked by socioeconomic inequalities and with a large part of the population that depends on its public health system (Unified Health System—SUS) [14], it is necessary to evaluate the factors associated to oral cancer under the view of their socioeconomic determinants and the population’s access to public health services. Thus, it is possible to develop evidence to encourage the implementation of public health policies to combat social inequalities and to strengthen the oral health service network in Brazil, and in similar populations.

Nonetheless individual factors such as sex, age, alcohol and tobacco consumption are remarkably associated to the incidence of oral cancer [7,8,9,10], other environmental socioeconomic variables should be investigated with regards their role on oral cancer incidence. Recent investigations from our research group have demonstrated the impact of socioeconomic factors and public oral health services on the rate of oral cancer hospitalization in Brazil [15,16]. Higher inequality rates, lower sanitation levels and poor education are associated with higher hospitalization rates [15]. In addition, higher availability of public oral health services contributed to lower oral cancer hospitalization rates and lower frequency of stage IV lesions [16].

However, a retrospective longitudinal analysis considering the variation on the oral cancer hospitalization rates within the last decades are still necessary. Also, the effect of both socioeconomic factors and public oral health services availability on the frequency of hospitalization and deaths due to oral cancer still need to be addressed. Studying factors associated to both hospitalization and mortality to oral cancer could help policy makers interfere more efficiently to reduce both morbidity and mortality of oral cancer in Brazil.

This study aimed to analyze the effect of socioeconomic variables and the influence of public oral health services availability on the frequency of hospitalization and deaths due to oral cancer in Brazil, under the macro-perspective of Brazilian cities.

## 2. Materials and Methods

An observational study was carried out with a retrospective cohort design. This study was based on a Brazilian information system that registers the cancer hospitalization information. Authors analyzed data regarding the perspective of the Brazilian cities. The sample units of the study were composed of Brazilian cities that registered, between 2002 and 2017, at least one hospitalized case of oral cancer (*n* = 4516).

The number of hospitalized cases of oral cancer was obtained from the Hospital Cancer Registry database of the National Cancer Institute (HRC-INCA–https://irhc.inca.gov.br/RHCNet/visualizaTabNetExterno.action) and was extracted according to the city of residence, for each year of the study. Only new cases of oral malignant tumors were considered, taking into account the year of the first diagnosis. Primary locations C00 to C06 (lip, base of tongue, tongue, gums, floor of mouth, palate, and other unspecified parts of the mouth) were considered. Final sample size consisted of 72,256 observations, between 2002 and 2017, within 4516 Brazilian cities, which resulted in 79,891 hospitalized cases of oral cancer. The number of deaths due to oral cancer primarily located on lip and oral cavity (C00 to C06), in the period between 2002 and 2017, was also collected for each city. The number of deaths consisted on the number of individuals who died within one year after the first treatment.

Data on total population, Municipal Human Development Index (MHDI) and Gini Coefficient were obtained from the panel of socioeconomic indicators of Brazilian cities, available in the Human Development Atlas of Brazil (http://atlasbrasil.org.br), linked to the United Nations Development Program (UNDP). The data were extracted with reference to the year 2010. The population size variable was distributed in the following categories: up to 30 thousand inhabitants, 30 to 50 thousand inhabitants, 50 to 100 thousand inhabitants and above 100 thousand inhabitants. The MHDI was classified as: low and very low (up to 0.599), medium (between 0.600 and 0.699), and high (equal to or greater than 0.700). The Gini coefficient was categorized as: less unequal (up to 0.4), and more unequal (above 0.4).

Data on oral health coverage in primary care and the number of Dental Specialized Center (DSC) were obtained from public reports from the “e-Gestor AB” portal, from the Ministry of Health’s Primary Care Secretariat, from 2002 to 2017 (https://egestorab.saude.gov.br/paginas/acessoPublico/relatorios/relatoriosPublicos.xhtml). Oral health coverage in primary care was categorized in up to 50% and above 50%. The cities were classified according to the presence or absence of DSC.

The data were organized and analyzed using the IBM Statistical Package for Social Sciences program (IBM SPSS, v. 24, IBM, Chicago, IL, USA). COX’s regression was used to determine the city’s risk ratio for having a new hospitalization due to oral cancer, between the years 2002 and 2017. The predictive variables of the model were: population size, MHDI, Gini coefficient, coverage of oral health in primary care and presence of DSC. The level of statistical significance was set at 5%. The model adjustment was assessed by the Omnibus test (*p* < 0.05). The effect of the predictor variables on the incidence of cases was verified by the Hazard Ratio (HR) measure, considering the 95% confidence interval. Accumulated risk curves were obtained for the adjusted model, as well as for each of the studied predictive variables.

In addition, the effect of socioeconomic factors and public oral health services availability on the number of deaths due to oral cancer was analyzed using multiple Poisson regression. The variable “presence of DSC” did not present statistical significance to be included in the model (*p* > 0.20). Therefore, the predictive variables of the model were: population size, MHDI, Gini coefficient and coverage of oral health in primary care. The level of statistical significance was set at 5%. The model adjustment was assessed by the Omnibus test (*p* < 0.05). The effect of the predictor variables on the incidence of cases was verified by the Incident Rate Ratio (IRR), considering the 95% confidence interval. A forest plot was used the graphically represent the magnitude of effect of each independent variable on the number of deaths due to oral cancer, according to the multiple Poisson regression model.

## 3. Results

Out of the 79,891 cases analyzed, 95.6% were of squamous cell carcinoma. Other malignant tumors consisted of mucoepidermoid carcinoma, adenocarcinoma and cystic adenoid carcinoma. The frequency of cases according to oral sites was the following: lip (9.6%), base of tongue (16.5%), tongue (26.5%), gums (3.2%), floor of mouth (13.0%), palate (14.8), and other unspecified parts of the mouth (16.4%).

Table 1 presents the descriptive data of the number of hospitalized cases of oral cancer in Brazil, according to the year, population size, municipal human development index, Gini coefficient, oral health coverage in primary care and the presence of Dental Specialties Centers. There is an increase in the average number of hospitalized cases of oral cancer in Brazil in the mid-2000s, followed by stability and a drop in the last years analyzed. The averages of the number of cases are higher in the municipalities with highest MHDI and Gini’s coefficient, oral health coverage in primary care up to 50% and with presence of DSC.

Cox’s multiple regression (Table 2) demonstrated that all variables that were studied had a significant effect on the number of hospitalized cases of oral cancer in Brazil. Cities above 50 thousand inhabitants, with high or very high MHDI, more unequal (Gini > 0.4), with less oral health coverage in primary care (<50%) and without the presence of DSC had a greater accumulated risk of having 1 or more hospitalized cases of oral cancer, in the period between 2002 and 2017. Cumulative risk curves for each variable under study are shown in Figure 1. Cumulative risk curves observed in Figure 1 reaffirm the statistical results reported in Table 2. Cities with more than 50 thousand inhabitants, with high or very high MHDI, with higher Gini index, and with lower coverage of public oral health services had all an increased cumulative risk for higher number of hospitalized cases of oral cancer.

The descriptive data regarding the number of deaths due to oral cancer, considering the period between 2002 and 2017, is presented in Table 3. Total number of deaths within the period under study consisted of 6735, which represents a mortality rate of 8.43%, within 4511 cities (Table 3). Higher mean values of deaths were detected in more populous, more developed and more unequal cities (Table 3). Cities with lower oral health coverage in primary care also presented higher mean number of deaths due to oral cancer (Table 3). Table 4 presents the multiple Poisson regression, which analyzed the effect of population size, MHDI, Gini Index and oral health coverage in primary care on the number of deaths due to oral cancer. Higher frequency of deaths is associated with higher population size, higher MHDI, higher Gini and lower oral health coverage in primary care (*p* < 0.001).

Figure 2 illustrates the magnitude of effect of each independent variable on the number of deaths due to oral cancer, according to the multiple Poisson regression model. The characteristic “Population size above 100 thousand inhabitants” presented the higher magnitude of effect. Overall, the model demonstrated that all variables under analysis resulted in a significative increase on the number of deaths (Figure 2).

## 4. Discussion

The findings of this investigation demonstrated that population size, socioeconomic status and availability of public oral health services influence the risk of hospitalization and the number of deaths due to oral cancer in Brazil. Two previous studies have developed similar analysis of oral cancer rates, involving socioeconomic characteristics and public oral health coverage in the SUS [13,15]. In these studies, different theoretical models were proposed, involving, for example, the work process in primary care and analysis by Brazilian regions and states. The present study considered a more comprehensive period and all cities of the country as sample units, which represented a higher sample size. Also, analysis on the factors associated to the number of deaths within this period contributes to the innovative perspective of this investigation.

It was shown that cities with large populations and high MHDI had a higher frequency of hospitalization and deaths due to oral cancer. In general, larger, and more developed cities have a better structured health service network, with greater encouragement and training of professionals for active disease tracking [17,18]. In addition, the population of those cities has higher life expectancy, which increases the proportion of the risk group for advanced age, in addition to greater access to diagnosis and hospitalization [19,20]. Also, it was expected that larger cities, with larger population, presented major number of cases [16]. Inverse associations between MHDI and oral cancer were found in studies that evaluated mortality rates [9,10,20]. These findings show that the largest number of hospitalized cancer cases is concentrated in more developed regions. Analyses on the number of deaths due to oral cancer revealed that socioeconomic factor that interfered with the frequency of oral cancer hospitalization, also influenced the frequency of deaths due to oral cancer. Once more developed and more populous areas may also have more inequality, this study reveals the impact of revealing the impact of social inequalities on the hospitalization and death due to oral cancer [21].

Nevertheless, results of the present study also demonstrated that cities with higher oral health coverage in primary care had lower number of hospitalization and lower number of deaths due to oral cancer. This corroborates a previous study from our research group, which highlighted the contribution of Brazilian National Oral Health Policy in reducing the frequency of hospitalization due to oral cancer [16]. In addition, investing in public oral health services may be an alternative to reduce social inequalities in health, and then also provide dignity and access to oral cancer prevention and treatment.

This analysis revealed that cities with strong inequality in the income distribution have a higher risk of presenting hospitalized cases of oral cancer. Socioeconomically disadvantaged individuals are often diagnosed with lesions in advanced clinical stages and with cervical metastases, which require complex treatments performed in the hospital setting [8,16]. Previous studies that used regions and states as sample units found no association between the Gini coefficient and oral cancer rates [13,17]. However, other studies with more specific and similar populations, considering cities and neighborhoods, demonstrated a positive association in this aspect, corroborating the present study [22,23]. So, this divergence of findings can be explained by the fact that only more homogeneous samples are able to detect a positive association between income inequality and oral cancer.

The average number of hospitalized cases of oral cancer increased as of 2004, coinciding with the implementation of the NOHP in Brazil. This can be explained by the increased frequency of diagnoses in SUS, as well as referral to hospital units and registration of cases in the information systems [24]. The scenario observed before this policy was characterized by a curative and individualized dental care model, centered on private services [25]. With its implementation, prevention started guiding oral health care, and procedures for detecting changes in the oral mucosa and biopsies started to be recommended in primary care and in the DSC [26].

The lower coverage of oral health in primary care and the absence of DSC are related to the higher risk of hospitalization and death due to oral cancer, according to the findings of this investigation. Studies have also found similar impacts of these variables on oral cancer mortality rates in Brazil in recent decades [13,16,17]. A recent investigation showed that the severity of hospitalized oral cancer cases was negatively associated with the expansion of the oral public health service network [16]. The lack of access to health services is one of the main factors related to the delay in the diagnosis of oral cancer, which often results in the need for more aggressive and mutilating treatments, reducing the individual’s survival [27].

Since the dentist in primary care acts as the entrance to the health system and the longitudinal character of this assistance in the SUS, it is able to develop educational actions to combat behavioral risk factors and self-detection of lesions through oral self-examination [28]. The DSC complement the assistance offered in primary care, representing the reference unit for suspected cases [29]. The relation between a greater number of DSC and a lower number of hospitalized cancer cases may indicate advances in terms of problem-solving, suggesting a higher frequency of cases detected in early stages, and reducing demand in hospital units. Results from this study shows that cities with lower oral health coverage in primary care may have higher number of deaths. This reaffirms the role of NOHP in contributing for the reducing the frequency of hospitalization and deaths due to oral cancer. Additional studies should be conducted to verify the effects of the organization of the oral cancer care network in the country.

Despite the advances observed, some barriers still make it difficult to address the issue of oral cancer in Brazil. Some problems, such as insufficient coverage, unequal distribution of health care units and a decrease in government investments in health in recent years, with the advance of austerity policies [14,17] are inherent to the health system as a whole. The lack of training and insecurity of professionals in relation to the diagnosis of malignant lesions and biopsies should also be highlighted [26]. Furthermore, the global Covid-19 pandemic in 2020, which required social distancing measures, as well as its long-term impacts can contribute to the increase of cases that are diagnosed later [30].

The present study has some limitations. The use of secondary data reduces the researchers’ control over the registration of information systems, which may represent a bias. Due to its observational design, this study suggests associations, but it is not the most appropriate for establishing cause and effect relationships. The study considered population data, and the phenomenon of ecological fallacy may occur if its findings are interpreted at individual levels. Oral cancer is a complex disease, strongly influenced by behavioral risk factors, which are not fully explained by individuals’ socioeconomic characteristics. Future studies must be developed considering these aspects as well. The use of official information systems, analysis at a city level throughout Brazil, the large sample size used, and the long period of analysis are strengths of this investigation.

Taking this into account, the expansion of oral health coverage in primary and specialized care, especially in the sense of equity, prioritizing populations in socioeconomic vulnerability, are essential strategies for improving the epidemiological scenario of oral cancer in Brazil [13,26]. In this sense, some urgent measures are highlighted, such as the increase in government investments in health, the implementation of public policies to combat social inequalities, the control of behavioral risk factors for oral cancer, the training of oral health professionals in the early identification of precancerous lesions and the expansion of health education measures for the general population.

This study highlights the relevance of socioeconomic factors and health services organization on the number of hospitalization and on the number of deaths due to oral cancer. This highlights that oral cancer is also social-determined disease. Based on that, a community-level approach to improve socioeconomic status and the availability of public health services may help to reduce the incidence of oral cancer. Future investigations should explore both the effect of individual and environmental variables on the incidence of hospitalization and deaths due to oral cancer.

## 5. Conclusions

The number of hospitalizations and deaths due to oral cancer in Brazil is influenced by cities’ population size, the population’s socioeconomic status and the availability of public dental services. Cities with 50,000 inhabitants or more, high MHDI, greater inequality in the distribution of income, with less oral health coverage in primary care and without the presence of a DSC had a greater cumulative risk of having 1 or more hospitalized cases of oral cancer. Higher frequency of deaths is associated with higher population size, higher MHDI, higher Gini and lower oral health coverage in primary care.

## Figures and Tables

**Figure 1 ijerph-18-00238-f001:**
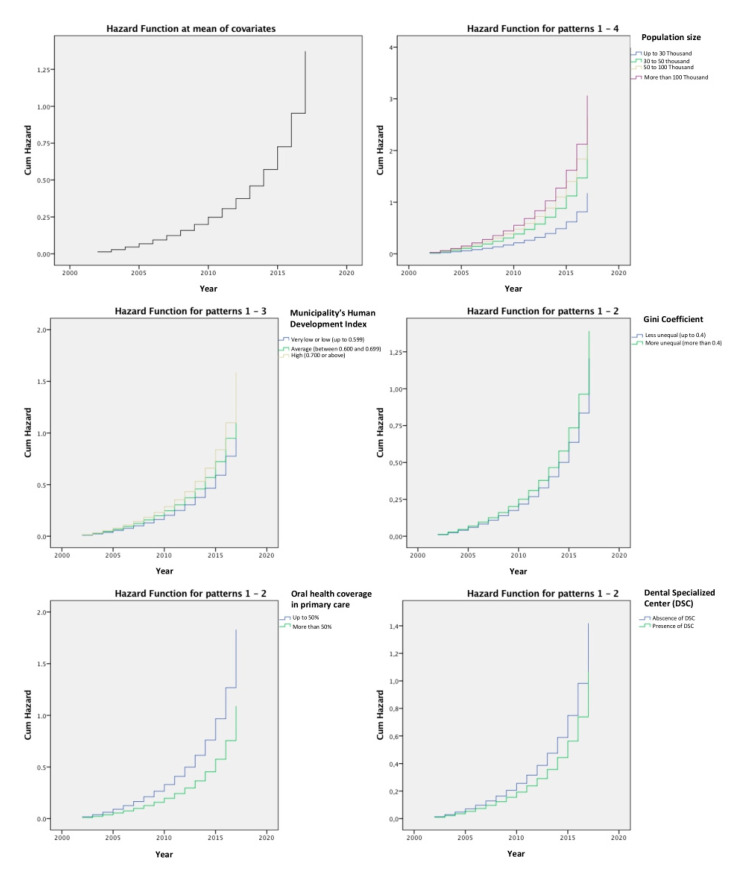
Cumulative risk curves of the mean effect of all variables (A) and for each category of each the independent variables.

**Figure 2 ijerph-18-00238-f002:**
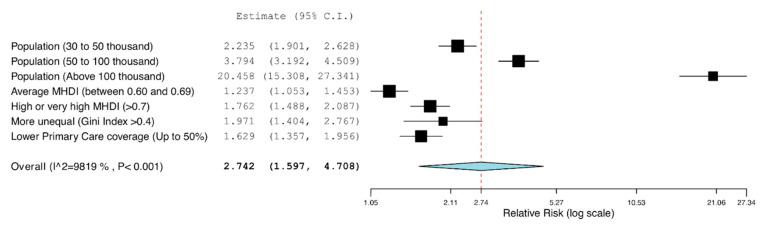
Forest plot that illustrates the magnitude of effect of each independent variable on the number of deaths due to oral cancer. Overall effect consists the average effect of all variables.

**Table 1 ijerph-18-00238-t001:** Descriptive statistics of hospitalized cases of oral cancer in Brazil, according to year, population size, Municipality’s Human Development Index (MHDI), Gini’s coefficient, Oral Health coverage in primary care, and Presence of Dental Specialized Center (DSC).

Variables		Hospitalized Cases of Oral Cancer						
		Mean	SD	Median	Max.	Min.	n	%
**Year**	2002	0.74	7.25	0	406	0	4516	6.3
	2003	0.75	6.73	0	344	0	4516	6.3
	2004	0.86	7.49	0	406	0	4516	6.3
	2005	0.96	7.37	0	398	0	4516	6.3
	2006	0.97	6.95	0	383	0	4516	6.3
	2007	1.06	7.07	0	366	0	4516	6.3
	2008	1.10	7.32	0	408	0	4516	6.3
	2009	1.20	8.05	0	436	0	4516	6.3
	2010	1.27	7.33	0	383	0	4516	6.3
	2011	1.33	8.52	0	477	0	4516	6.3
	2012	1.29	8.41	0	484	0	4516	6.3
	2013	1.35	8.27	0	474	0	4516	6.3
	2014	1.37	8.60	0	478	0	4516	6.3
	2015	1.37	7.37	0	394	0	4516	6.3
	2016	1.16	4.62	0	192	0	4516	6.3
	2017	0.90	3.81	0	149	0	4516	6.3
**Population Size**	Up to 30 thousand inhabitants	0.38	0.75	0	30	0	57,088	79.0
	30 to 50 thousand inhabitants	1.09	1.41	1	9	0	6496	9.0
	50 to 100 thousand inhabitants	1.99	2.13	1	19	0	4704	6.5
	Above 100 thousand inhabitants	10.54	29.31	5	484	0	3968	5.5
**Municipality’s Human Development Index (MHDI)**	Low or very low MHDI (up to 0.59)	0.33	0.67	0	7	0	17,520	24.2
	Average MHDI (between 0.60 and 0.69)	0.54	1.18	0	30	0	28,608	39.6
	High or very high MHDI (above 0.7)	2.24	12.00	0	484	0	26,128	36.2
**Gini’s coefficient**	Less unequal (<0.4)	0.48	1.04	0	19	0	5968	8.3
	More unequal (>0.4)	1.16	7.62	0	484	0	66,288	91.7
**Oral Health coverage in primary care**	Up to 50%	1.82	10.78	0	484	0	32,333	44.7
	Above 50%	0.53	1.37	0	50	0	39,923	55.3
**Presence of Dental Specialized Center (DSC)**	Without DSC	0.63	2.97	0	406	0	64,304	89.0
	With DSC	4.96	19.94	1	484	0	7952	11.0

SD: standard deviation, Max.: Maximum value, Min.: Minimum value, n: absolute frequency, %: relative frequency.

**Table 2 ijerph-18-00238-t002:** Multiple Cox regression that determined cumulative risk of each explanatory variable on the frequency of hospitalized cases of oral cancer in Brazil, between 2002 and 2017.

	B	*p*-Value	HR	95% CI	
				Lower	Upper
Population size		<0.001			
Up to 30 thousand inhabitants			Reference		
30 to 50 thousand inhabitants	0.000	0.998	1.000	0.972	1.029
50 to 100 thousand inhabitants	0.223	<0.001	1.250	1.215	1.287
Above 100 thousand inhabitants	0.369	<0.001	1.446	1.400	1.492
Municipality’s Human Development Index (MHDI)		<0.001			
Low or very low MHDI			Reference		
Average MHDI	0.017	0.064	1.017	0.999	1.036
High or very high MHDI	0.166	<0.001	1.180	1.158	1.203
Gini’s coefficient		<0.001			
Less unequal (<0.4)			Reference		
More unequal (>0.4)	0.071	<0.001	1.074	1.048	1.101
Oral Health coverage in primary care		<0.001			
Up to 50%	0.260	<0.001	1.296	1.278	1.315
Above 50%			Reference		
Presence of Dental Specialized Center (DSC)		<0.001			
Without DSC	0.142	<0.001	1.153	1.132	1.174
With DSC			Reference		

B: regression coefficient, *p*-value: statistical significance, HR: hazard ratio, 95% CI: 95% confidence interval.

**Table 3 ijerph-18-00238-t003:** Descriptive statistics of the deaths due to oral cancer in Brazil, according population size, Municipality’s Human Development Index (MHDI), Gini’s coefficient and Oral Health coverage in primary care, within the period between 2002 and 2017.

Variables		Number of Deaths due to Oral Cancer						
		Mean	SD	Median	Max.	Min.	n	%
**Population Size**	Up to 30 thousand inhabitants	0.51	1.01	0	11.00	0	3563	79.0
	30 to 50 thousand inhabitants	1.35	1.95	1.00	10.00	0	406	9.0
	50 to 100 thousand inhabitants	2.41	3.23	1.00	17.00	0	294	6.5
	Above 100 thousand inhabitants	14.75	40.82	5.00	484.00	0	248	55.0
**Municipality’s Human Development Index (MHDI)**	Low or very low MHDI (up to 0.59)	0.51	0.96	0	11.00	0	1092	24.2
	Average MHDI (between 0.60 and 0.69)	0.88	1.95	0	28.00	0	1787	39.6
	High or very high MHDI (above 0.7)	2.82	16.70	0	484.00	0	1632	36.2
**Gini’s coefficient**	Less unequal (<0.4)	0.45	1.24	0	13.00	0	373	8.3
	More unequal (>0.4)	1.59	10.62	0	484.00	0	4138	91.7
**Oral Health coverage in primary care**	Up to 50%	1.70	11.04	0	484.00	0	3821	84.7
	Above 50%	0.34	0.88	0	11.00	0	690	15.3

SD: standard deviation, Max.: Maximum value, Min.: Minimum value, n: absolute frequency of cities, %: relative frequency.

**Table 4 ijerph-18-00238-t004:** Multiple Poisson regression that determined rate ratio of each explanatory variable on the frequency of deaths due to oral cancer in Brazil, between 2002 and 2017.

	B	*p*-Value	IRR	95% CI	
				Lower	Upper
Population size		<0.001			
Up to 30 thousand inhabitants			Reference		
30 to 50 thousand inhabitants	0.804	<0.001	2.235	1.900	2.628
50 to 100 thousand inhabitants	1.333	<0.001	3.794	3.192	4.509
Above 100 thousand inhabitants	3.018	<0.001	20.458	15.308	27.341
Municipality’s Human Development Index (MHDI)		<0.001			
Low or very low MHDI			Reference		
Average MHDI	0.212	0.010	1.237	1.052	1.453
High or very high MHDI	0.567	<0.001	1.762	1.488	2.087
Gini’s coefficient		<0.001			
Less unequal (<0.4)			Reference		
More unequal (>0.4)	0.679	<0.001	1.971	1.405	2.767
Oral Health coverage in primary care		<0.001			
Up to 50%	0.488	<0.001	1.629	1.357	1.956
Above 50%			Reference		

B: regression coefficient, *p*-value: statistical significance, IRR: incident risk ratio, 95% CI: 95% confidence interval.

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
