# Peer review of "Influence of Public Oral Health Services and Socioeconomic Indicators on the Frequency of Hospitalization and Deaths due to Oral Cancer in Brazil, between 2002–2017"

_ijerph, 2020, doi:10.3390/ijerph18010238_

Round 1

Reviewer 1 Report

Dear Authors

the article is interesting and well written. Only minr changes are necessary.

Dear Authors

The article is interesting and well written Only minor concerns have to be addressed:

  • The title should consider the years that were used for the present investigation (2002-2017).

ABSTRACT 

  • Subtitles should be considered: aim, materials and methods, results and conclusions.
  • The content of the conclusion should be considered as correctly described in point 5 (conclusions) of this article.

RESULTS

  • According to the article: “The average number of hospitalized cases of oral cancer increased as of 2004, coinciding with the implementation of the NOHP in Brazil. This relation is explained by the increase in the frequency of diagnoses in the SUS, as well as referral to hospital units and registration of cases in the information  systems [22]. “ but with the implementation of NOHP (National Oral Health Policy ), the cases of cancer should decrease as mentioned in this part of the article.

Kind regards

Author Response

We are very thankful for the careful review of the text and we seek to comply with the recommendations as described below. Changes to the text are marked in red.

Reviewer 1:

Dear Authors

The article is interesting and well written. Only minor concerns have to be addressed.

The title should consider the years that were used for the present investigation (2002-2017).

Answer: Thank you for the observation. We added the information to the title.

ABSTRACT 

Subtitles should be considered: aim, materials and methods, results and conclusions.

Answer: Thank you for the observation. We added the subtitles to the abstract.

The content of the conclusion should be considered as correctly described in point 5 (conclusions) of this article.

Answer: Thank you for the observation. We added to the abstract the same content as the conclusions of the article, considering the word limit.

RESULTS

According to the article: “The average number of hospitalized cases of oral cancer increased as of 2004, coinciding with the implementation of the NOHP in Brazil. This relation is explained by the increase in the frequency of diagnoses in the SUS, as well as referral to hospital units and registration of cases in the information systems [22]. “ but with the implementation of NOHP (National Oral Health Policy ), the cases of cancer should decrease as mentioned in this part of the article.

Answer: Thank you for the observation. Oral cancer is a complex disease, with cumulative etiological factors, and a long-term manifestation. The implementation of prevention and control policies (such as NOHP) contributes to an increase in the frequency of diagnoses and referrals to hospital units, as they prioritize the active screening of the disease (lines 152-154, 155-157). Thus, an initial increase in the number of cases is expected, followed by a reduction in the future, as a result of effective preventive measures. In addition, a recent investigation (Raymundo et al., 2020) showed that the severity of hospitalized oral cancer cases was nega­tively associated with the expansion of the oral public health service network. This reference was added to the text.

Authors also highlight the English language was revised. In addition, authors improved the description of design, methods and results sections.

Reviewer 2 Report

Oral cancer, as the most challenging pathological disease in maxillofacial region, is a very meaningful topic to study on. It has been well acknowledged that socioeconomics of patients influence many oral cancers, such as OSCC, HSV, melanoma, etc. This study investigated the associations of oral cancer cases with social economic conditions of patients in Brazil at a national level, which is very valuable in aspect of public health data and health care promotion. However, authors need to cherish such substantial amount of data and deeply investigated how this study would enhance the knowledge of understanding of oral cancer. Current version of the paper simplify laid out the raw data without associating data with clinical guidance or general understanding of oral cancers. The manuscript needs significant revision to improve its impact in oral cancer field.

Here are some recommendations.

  1. Highly recommend to collect the motility rate of current samples, and analyze the associations of parameters with motility rate in a national level. The data will be way more valuable than current plain distribution of cases.
  2. Associate social economics with specific oral cancer diseases at a national level.
  3. Methodology:

Add distribution of cancer diagnosis in current samples. Though 98% are OSCC, how many cancers are included in this study? What is the definition of cancer in this study? Is benign oral tumor regarded as oral cancer in this study? Locations of oral cancer samples should be concluded with a distributional ratio as well.

  1. Results:

Results need major revisions. Despite that raw data were listed in charts, conclusive descriptions of your data should be presented in result section.  Data should be concluded by the authors rather than the readers.

  1. Discussions:

Recommend re-written of the discussion part. Overall, discussion used multiple other studies to present a meaningful influence of current study, however, current study provided little data or evidence to relate these citations with its own data.

Here are a few examples.

Line 119-121. Quote “Two previous studies have developed similar analyses of oral cancer rates, involving socioeconomic characteristics and dental coverage in the SUS among the regions and states of Brazil [13,15].

Please address more details of pros and cons of current study compared with previous similar studies. Please also compare the data and conclusions to reveal the innovative information discovered in the current study.

Line 124 to 131. Current study found that population sizes in different cities associate with the frequency of hospitalized cases.  The explanations regarding oral cancer distribution and motility after this result are very meaningful, however, current study should provide its own data to support these statements. Frequency alone is not strong enough to extend into following statements. As a result, frequency alone in this conclusion provides little value to oral cancer in the level of public health.

Line 135-139

What might be the reasons to cause the complete opposite results of the associations of Gini coefficient and oral cancer rates between studies of #13-14 and studies of # 20-21 & current study in the same country? Please analyze possible reasons.

Author Response

Reviewer 2:

Oral cancer, as the most challenging pathological disease in maxillofacial region, is a very meaningful topic to study on. It has been well acknowledged that socioeconomics of patients influence many oral cancers, such as OSCC, HSV, melanoma, etc. This study investigated the associations of oral cancer cases with social economic conditions of patients in Brazil at a national level, which is very valuable in aspect of public health data and health care promotion. However, authors need to cherish such substantial amount of data and deeply investigated how this study would enhance the knowledge of understanding of oral cancer. Current version of the paper simplify laid out the raw data without associating data with clinical guidance or general understanding of oral cancers. The manuscript needs significant revision to improve its impact in oral cancer field.

Here are some recommendations.

Answer: Thank you very much for your comments. We exerted our best efforts to carry out the suggestions.

Highly recommend to collect the motility rate of current samples, and analyze the associations of parameters with motility rate in a national level. The data will be way more valuable than current plain distribution of cases.

Answer: Thank you for the suggestion. We agree that the use of mortality data would be very valuable in this investigation. The database is very limited with regards information on the number of deaths. However, we exerted our best efforts to consider your suggestion and improve the quality of the manuscript.

For our data, the death due to oral cancer was defined as the number of individuals who died within one year after the first treatment. For this study, we can analyze the absolute frequency of deaths for each city included in the study. Total number of deaths within the period under study consisted of 6,735, which represents a mortality rate of 8.43%. In addition to this, we analyzed factors that are associated with the number of deaths.

Methodology:

Add distribution of cancer diagnosis in current samples. Though 98% are OSCC, how many cancers are included in this study? What is the definition of cancer in this study? Is benign oral tumor regarded as oral cancer in this study? Locations of oral cancer samples should be concluded with a distributional ratio as well.

Answer: Thank you very much for your comments. We added the frequency distribution according to the independent variables under study. The final sample consisted of 72,256 observations, between 2002 and 2017, within 4516 cities, which resulted in 79891 hospitalized cases of oral cancer. For this study, only malignant tumors were considered for analysis. In addition to oral squamous cell carcinoma, other malignant tumors consisted of mucoepidermoid carcinoma, adenocarcinoma and cystic adenoid carcinoma. We also included the frequency of cases according to the primary location of oral cancer.

Results:

Results need major revisions. Despite that raw data were listed in charts, conclusive descriptions of your data should be presented in result section.  Data should be concluded by the authors rather than the readers.

Answer: Thank you for the suggestion. We added conclusive descriptions within the Results section.

Discussions:

Recommend re-written of the discussion part. Overall, discussion used multiple other studies to present a meaningful influence of current study, however, current study provided little data or evidence to relate these citations with its own data.

Here are a few examples.

Line 119-121. Quote “Two previous studies have developed similar analyses of oral cancer rates, involving socioeconomic characteristics and dental coverage in the SUS among the regions and states of Brazil [13,15].

Please address more details of pros and cons of current study compared with previous similar studies. Please also compare the data and conclusions to reveal the innovative information discovered in the current study.

Answer: Thank you for the observation. The comparison of the main results of the similar studies cited with those of the current study is present throughout the discussion (lines 145-147, 159-161). We rewrote the excerpt, adding details that differentiate the previous studies from the current one, highlighting its innovative character (lines 125-131).

Line 124 to 131. Current study found that population sizes in different cities associate with the frequency of hospitalized cases.  The explanations regarding oral cancer distribution and motility after this result are very meaningful, however, current study should provide its own data to support these statements. Frequency alone is not strong enough to extend into following statements. As a result, frequency alone in this conclusion provides little value to oral cancer in the level of public health.

Answer: Thank you for the observation. We believe that the Cox’s multivariate regression results support this information. Our results showed that cities with a larger population had a greater accumulated risk of having 1 or more hospitalized cases of oral cancer. In addition, we analyzed the frequency of deaths due to oral cancer, considering period of one year after the first treatment. This is the best data on mortality that the dataset can provide. We found that higher frequency of deaths is associated with higher population size, higher HDI, higher Gini and lower oral health coverage in primary care (p<0.001). This analysis was included in the text and appropriate discussion was inserted within the text.

Line 135-139

What might be the reasons to cause the complete opposite results of the associations of Gini coefficient and oral cancer rates between studies of #13-14 and studies of # 20-21 & current study in the same country? Please analyze possible reasons.

Answer: Thank you for the observation. As discussed, studies #13-14 used regions and states as sample units, which represent broader and more heterogeneous populations. Studies #22-23 and the present study used populations more similar to each other (cities) and more homogeneous samples, possibly being able to detect an association between Gini coefficient and oral cancer rates. We rewrote the passage to facilitate the understanding of this interpretation.

Reviewer 3 Report

The study is interesting and was carried out with appropriate methodology and with scientific rigor. The design of the study is correct.

Only a few considerations need to be made to improve the manuscript:

1) The background to the study needs to be expanded.

2) Authors should check the bibliographical references.

I consider this manuscript to be suitable for publication in the IJERPH journal.

The manuscript is original and interesting for the readers.

The methodology used in the research project is appropriate.

The results described are consistent and the conclusions correspond to the objectives proposed in the manuscript.

For the publication of the manuscript I consider that the authors should check the bibliographical references. Bibliographic references are not described according to the journal's guidelines. Authors should also expand on the background of the study in the introduction to the manuscript.

Author Response

The study is interesting and was carried out with appropriate methodology and with scientific rigor. The design of the study is correct.

Only a few considerations need to be made to improve the manuscript:

1) The background to the study needs to be expanded.

Answer: Many thanks you for the observation. We included a new paragraph at the end of Introduction section to better support the rationality of the study. Two new references were added to the text.

2) Authors should check the bibliographical references.

Answer: Thank you for the observation. We made the necessary adjustments to the formatting of the references according to the journal's guidelines.

We also revised the text according to English language and style.

Round 2

Reviewer 2 Report

To authors:

The author team made efficient revisions based on suggestions. The manuscript consists of much more information than the previous version. The discussion is well updated.

Here are some more suggestions:

  1. English revision: examples: Line 115, line 241, correct spelling
  2. Line 161-165, add table 3 as the reference
  3. Table 1, Table 2, Table 3, and table 4: please keep the descriptive tables, and add charts/figures to show the associations of variables and results and the trending (line 131-132) of the change more straightforwardly, such as in figure 1.

Overall, it is a meaningful study, however, no innovative input was provided by this study. The influence of this study on public health, in general, is questionable. The influence of this study on local oral cancer research in Brazil is solid.

Author Response

We are very thankful for the careful review of the text and we seek to comply with the recommendations as described below. Changes to the text are marked in red.

Reviewer 1:

The author team made efficient revisions based on suggestions. The manuscript consists of much more information than the previous version. The discussion is well updated.

Answer: Thanks for your comments.

Here are some more suggestions:

English revision: examples: Line 115, line 241, correct spelling

Answer: Thank you for the observation. We corrected the sentences.

Line 161-165, add table 3 as the reference

Answer: Thank you for the observation. Table 3 is referenced in lines 172-173.

Table 1, Table 2, Table 3, and table 4: please keep the descriptive tables, and add charts/figures to show the associations of variables and results and the trending (line 131-132) of the change more straightforwardly, such as in figure 1.

Answer: Thank you for your comment. Appropriate changes were made within the text and a new figure was added at the end of Results section to illustrate results of table 4.

Overall, it is a meaningful study, however, no innovative input was provided by this study. The influence of this study on public health, in general, is questionable. The influence of this study on local oral cancer research in Brazil is solid.

Answer: Thank you for your comment. The innovative aspect of the study was highlighted at the end of Introduction and Discussion sections.